# A Novel Hybrid Interval Rough SWARA–Interval Rough ARAS Model for Evaluation Strategies of Cleaner Production

**Ilija Tanackov** [1], **Ibrahim Badi** [2], **Željko Stević** [3,*], **Dragan Pamučar** [4], **Edmundas Kazimieras Zavadskas** [5] **and Romualdas Bausys** [6]

1   Faculty of Technical Sciences, University of Novi Sad, Trg Dositeja Obradovića 6, 21000 Novi Sad, Serbia; ilijat@uns.ac.rs
2   Libyan Academy, Qasr Ahmed Street, Misurata 2449, Libya; i.badi@lam.edu.ly
3   Faculty of Transport and Traffic Engineering, University of East Sarajevo, Vojvode Mišića 52, 74000 Doboj, Bosnia and Herzegovina
4   Department of Logistics, University of Defence, Belgrade, Pavla Jurišića Šturma 33, 11000 Belgrade, Serbia; dpamucar@gmail.com
5   Institute of Sustainable Construction, Faculty of Civil Engineering, Vilnius Gediminas Technical University, Sauletekio al. 11, LT-10223 Vilnius, Lithuania; edmundas.zavadskas@vilniustech.lt
6   Department of Graphical Systems, Vilnius Gediminas Technical University, Sauletekio al. 11, LT-10223 Vilnius, Lithuania; romualdas.bausys@vilniustech.lt
*   Correspondence: zeljkostevic88@yahoo.com or zeljko.stevic@sf.ues.rs.ba

**Abstract:** Cleaner production is certainly a challenge of our everyday life, and a lot of effort and energy is required to achieve it. This paper has created a model of five strategies for cleaner production in Libyan industry, which have been evaluated on the basis of eight criteria. In order to determine the significance of the criteria, a novel interval rough SWARA (step-wise weight assessment ratio analysis) method has been developed, which takes into account the preferences of decision-makers (DMs) by applying interval rough numbers. A novel interval rough ARAS (additive ratio assessment) method has been developed for the evaluation and selection of the most favorable strategy for cleaner production. The integration of the developed methods has yielded results showing that the first strategy, launching awareness-raising campaigns to publicize these policies, represents the most realistic and best current solution to achieve cleaner production in Libyan industry. A comparative analysis with some existing interval rough methodologies has been presented to verify the superiority of the proposed model. In addition, in a sensitivity analysis, the weight of the most significant criterion has been changed.

**Keywords:** cleaner production; interval rough numbers; SWARA; ARAS; multi-criteria decision-making

## 1. Introduction

Presently, it has become necessary to pay attention to the various effects of industrial activity on the environment, particularly considering the advanced state of degradation that the planet has reached. The irrational exploitation of resources and the exposure of some of them to depletion, together with various forms of pollution, are threatening the security of human beings and others and make the future of the coming generations unstable. On the other hand, industrial institutions cannot carry out their activities without affecting or being affected by the environment, as the latter is both a source for its inputs and a dumping ground for its waste. Although the industrial sector is the main driving force of countries' economies [1], it is one of the most threatening elements to the environment due to the production patterns used. It is, therefore, necessary to reflect on how to reduce these threats without damaging the economic output of industrial institutions, because this determines one of the most critical factors impacting their continuity.

Cleaner production requires the integration of the three dimensions of sustainable development into the production process, both before and after as well as throughout

the product life cycle, through the adoption of technologies that take these dimensions into account, all in the presence of environmental management processes that prepare the respective environment for the application of the technologies [2]. In support of the implementation of this concept, several initiatives have emerged at international organizations such as the United Nations Industrial Development Organization (UNIDO) and the United Nations Environment Program (UNEP), which have adopted several programs including the National Cleaner Production Centres (NCP) [3].

To achieve the best-desired effect, cleaner production practices should be permanently integrated throughout the organization and managed openly through effective leadership and communication [4]. These practices need to be implemented strategically, including in operations, products, and services, to optimize the use of natural resources and to reduce the quantity of waste and pollutant emissions, which pose a risk to human safety and health [5]. In turn, this will increase competitiveness by increasing revenue and reducing waste [6].

Industrial institutions are currently operating in a dynamic and rapidly changing environment. As a result of the increased attention on environmental problems and the inclusion of the environmental aspect among the most recent areas of competition, these institutions have moved towards adopting strategies that include the environmental dimension in their priorities in order to maintain and develop their competitive position, thus enabling them to ensure their continuity [7]. Studies have shown that the adoption of cleaner production policies in the manufacturing sector has had positive effects on the environment and plant workers [8,9]. The implementation of these policies has proven to be effective for economic development and the environment. Various cleaner production policies can be realized within the framework of sustainability methods. For dealing with the sustainability issue, the hybrid multi-criteria decision methods are of the utmost importance since they provide the possibility to consider interrelated criteria and to study the relative importance of each separate criterion [10].

The most important objectives that can be achieved by further research are the following:

(1) to provide the best possible basis for the application of cleaner production in Libyan industry,
(2) the development of a novel integrated interval rough SWARA–interval rough ARAS taking into account uncertainties in decision making, and
(3) the additional enrichment of rough set theory and MCDM areas through the development and application of this integrated model.

In addition to presenting the need for research, motives, and goals, this paper is structured through five other sections. Section 2 shows the current situation in Libya in terms of cleaner production. Section 3 presents the developed model. First, the basic operations with interval rough numbers are given, and then the development of novel IRN SWARA and IRN ARAS methods is explained in detail. Section 4 presents the evaluation of strategies for cleaner production. First, a model setting is given with explanations of the criteria and potential strategies, followed by a detailed calculation for both developed methods. Section 5 consists of the verification of the results through a sensitivity analysis and a discussion, while the Section 6 presents the conclusion.

## 2. Literature Review

### 2.1. MCDM Applications in Cleaner Production

Cleaner production can be defined as an integrated preventive strategy that can be applied to industrial processes and end products to improve economic efficiency and reduce environmental risks. Cleaner production includes reducing waste and emissions at the source and conserving raw materials and energy. Decision makers are faced with many complex decisions at different levels related to the implementation of this strategy. These decisions involve several criteria that must be taken into account. From this point of view, multi-criteria methods have been used in different applications within the cleaner production concept. For instance, Nikolić et. al. developed a multi-criteria model to select the optimal technology for copper smelting, where the PROMETHEE/GAIA methodology

was used for the multi-criteria analysis [11]. The available copper smelting techniques applied under different production conditions were ranked. Govindan et. al. studied the main obstacle to the remanufacturing of automotive parts in India, whereby a model was proposed and applied to an industrial case [12]. Two methodologies were used as part of the solution: Interpretive Structured Modeling (ISM) and the Analytical Network Process (ANP). The paper revealed that a high cost and the non-acceptance of customers are the main and most influential barriers to remanufacturing of auto parts in India. Liang et. Al. proposed a model for the evaluation of cleaner production for gold mines [13]. A set of evaluation criteria was proposed, and a modified methodology was used to rank expert opinions based on (PLTs) for the purpose of calculating the weights of the sub-criteria, while using the (TODIM) method for the purpose of ranking the proposed alternatives. A study by Promentilla et. al. proposed a decision model using the analytical hierarchical network process to address the complexity and ambiguity involved in the selection of clean technologies, where the problem was analyzed in a hierarchical network structure, and the probability distribution of weights required for the arrangement was derived [14]. A model was developed by Zhang and Haapala to evaluate impacts on sustainability by performing economic evaluations and environmental and social impact evaluations, where the evaluation results were integrated into the sustainable manufacturing evaluation framework, together with a modified weighting method based on pairwise comparison and the higher order decision making method [15]. Research work was conducted on a manufacturing case in the stainless-steel knives industry, and the study revealed that the cost of the cutting tool is the largest contributor to production costs. Tseng et al. also formulated the problem of selecting competitive priorities for a PCB manufacturer [16]. Their study addresses important decision criteria for implementing cleaner production, such as: organization and techniques, evaluation and feedback, training, and the people involved in determining the best competitive priorities. The Fuzzy Analytical Network Process (FANP) method was used to determine the final priority of various decision criteria.

### 2.2. Cleaner Production in Libya

Libya invested in the industrial sector, whether in small, medium, or large industries, during the period from the 1970s until the beginning of the new millennium [17,18]. Similarly, the private industrial sector has been active over the last ten years [19]. Not all of these investments have been accompanied by a similar contribution to the domestic product (DP), nor by clear environmental strategies, resulting in the emergence of many environmental problems [20]. The Libyan legislature may have dealt with the environmental aspects when enacting many laws, but these laws need to account for everything related to cleaner production. At the level of practices, it does not seem that companies, whether in the public or private sector, are putting environmental aspects at the forefront of their attention, either due to this gap in regulatory laws or insufficient monitoring, or because they struggle to compete on the local market with products imported from abroad [21]. Hence, the idea of identifying the most critical obstacles to the implementation of cleaner production policies has emerged from this research, as well as the appropriate strategies to overcome these obstacles.

The Libyan state has endeavored to increase the industrial sector's contribution to the gross domestic product (GDP) in a manner that leads to a reduction in the share of the oil sector. To achieve this goal, the state has worked on implementing numerous development plans and programs in which the industrial sector has had a share in establishing many industrial projects, and the total investments in this sector during the period from 1995 to 2006 reached more than 13,000 million Libyan dinars [22]. On the other hand, the contribution of this sector to GDP remained low and did not exceed 5%. Figure 1 shows the contribution of the different sectors to Libya's GDP. Fluctuating oil prices may have been one of the reasons for the change in expenditure from one period to another. Over the past ten years, many small and medium private industrial projects were implemented, but it is difficult to obtain accurate statistics on their number or the value of the investments

made in them. On the other hand, most public plants are at a standstill due to aging or the inability to compete with products supplied from abroad, with the exception of large industries such as the steel and cement industries.

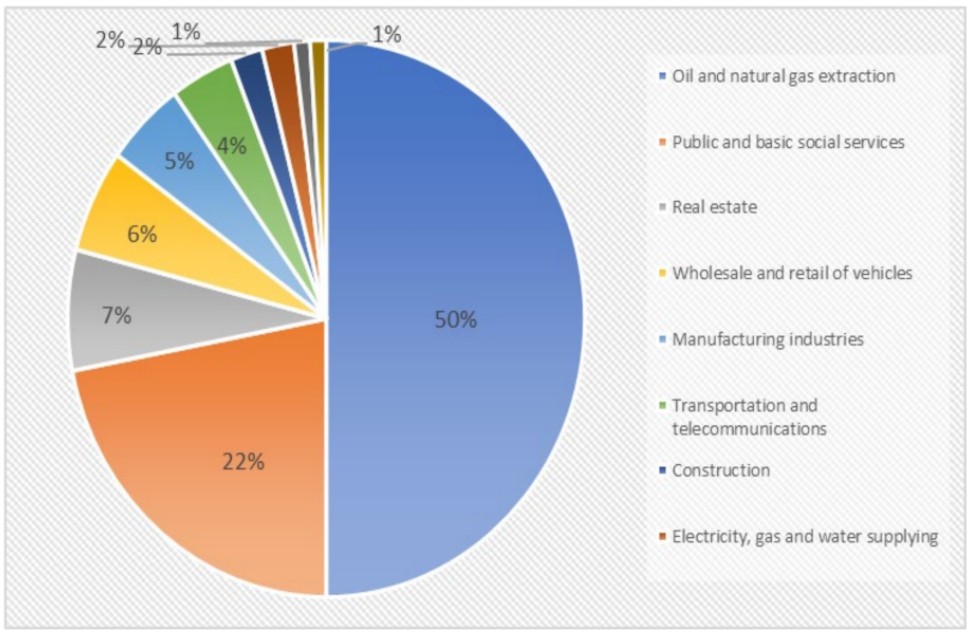

**Figure 1.** Contribution of the different sectors to Libya's GDP (adopted from [23]).

Since the beginning of 2011, the industrial plan, which was scheduled to last three years from 2010 up to 2013 and was expected to have a total investment value of 3.39 billion dinars and include about 2191 new projects, has been stalled due to the unstable situation in the country. In terms of the industrial sector's contribution to GDP, the latter was estimated at 324,020 million LYD during the period from 2010 to 2014, while the oil sector accounted for 265,703 million LYD and 82% of GDP, and the manufacturing sector represented 9421 LYD and only 3% of GDP. Conversely, loans to the industrial sector from the Development Bank-Libya during the period from 2011 to 2015, for financing the operation or development of industrial activities, amounted to 39.4 million dinars, whereas loans from the same bank to finance the same sector reached 379.7 million dinars for the period from 2007 to 2010 [24].

In terms of cleaner production, Libya's experience is still very weak. For example, although Libyan laws are concerned about the environment in some respects, there is no law on cleaner production. Environmental impact assessments (EIA) are still not taken into account, whether in existing or contracted projects. The most important laws concerning the environment can be summarized as follows [20,25]:

a law enacted in 1989 concerning regulatory procedures in the industrial sector (Law No. 22);

a law enacted in 1982 concerning environmental legislation (Law No. 7);

a law enacted in 1991 on industrial wastewater (Law No. 13);

Law No. 5 of 1990, on standardization and metrology.

These laws are used as an incentive to stimulate cleaner production. At the level of organizations with interest in the environment, the most important of these laws can be explained by the organizations in the following list [25]:

- The Centre for Industrial Research.
- Libyan National Centre for Standardisation and Metrology (LNCSM).
- The Environment General Authority (EGA).

### 3. Methods

In this section of the paper, a new methodology applied for the evaluation of cleaner production strategies is presented. First, basic operations with IRNs are given in order to facilitate understanding and present further observations of the paper, i.e., calculations presented for a specific example in Libya. Then, as a great contribution of the paper, the extension of previously developed MCDM methods is presented [26]: SWARA and ARAS with interval rough numbers.

The main advantages that determined the authors' use of the SWARA method are as follows: (1) The SWARA method has a simple and easy-to-understand mathematical apparatus. It can therefore be used effectively in group decision making processes in which experts can easily exchange views; (2) The complexity of the mathematical apparatus of the SWARA method does not increase with the increase in the number of criteria. Therefore, it is suitable for application in complex mathematical models that have a number of decision attributes; and (3) the SWARA method allows the processing of data defined on the basis of different assessment scales. This feature gives this method significant flexibility, allowing for the easy adaptation of the SWARA model to specific situations.

The ARAS method has proven to be a stable and reliable tool for decision making in a dynamic environment. Compared to other traditional MCDM techniques. The ARAS method is significantly resistant to the rank reversal problem. If an ideal and anti-ideal value is defined outside the range of home matrix values, the ARAS methodology algorithm eliminates the rank reversal problem. Numerous simulations in which larger data sets are used have also shown the stability of the ARAS methodology, which supports its application in more complex studies in which larger data sets are processed in a short time.

#### 3.1. Operations with Interval Rough Numbers

Rough numbers have recently emerged as a new concept capable of dealing with uncertain, ambiguous, and probabilistic data [27,28]. This ability to deal with ambiguous and imprecise data has prompted many researchers to use these numbers in many applications and problems faced by decision makers and researchers in the real world [29]. In the same context, this paper proposes a model for the extension of interval rough numbers. The application of interval rough numbers in the decision-making process enables the efficient processing of imprecision and uncertainty in subjective estimates. The rough boundary interval is an imprint of uncertainty defined based on internal knowledge in the home matrix data. In cases where greater uncertainties are present in the data, the footprint of uncertainty increases, while in cases where imprecision and uncertainty from the data are eliminated, the rough number interval is transformed into a crisp value. The application of interval rough numbers eliminates the need for additional subjective assessments of boundary parameters and significantly objectifies the decision-making process.

IRNs are described by specific arithmetic operations. Arithmetic operations between two interval rough numbers $IRN(A) = ([a_1, a_2], [a_3, a_4])$ and $IRN(B) = ([b_1, b_2], [b_3, b_4])$ are performed using Expressions (1)–(5) [30]:

(1) Addition of interval rough numbers, "+",

$$IRN(A) + IRN(B) = ([a_1, a_2], [a_3, a_4]) + ([b_1, b_2], [b_3, b_4]) = ([a_1 + b_1, a_2 + b_2], [a_3 + b_3, a_4 + b_4]) \tag{1}$$

(2) Subtraction of interval rough numbers, "-",

$$IRN(A) - IRN(B) = ([a_1, a_2], [a_3, a_4]) - ([b_1, b_2], [b_3, b_4]) = ([a_1 - b_4, a_2 - b_3], [a_3 - b_2, a_4 - b_1]) \tag{2}$$

(3) Multiplication of interval rough numbers, "×",

$$IRN(A) \times IRN(B) = ([a_1, a_2], [a_3, a_4]) \times ([b_1, b_2], [b_3, b_4]) = ([a_1 \times b_1, a_2 \times b_2], [a_3 \times b_3, a_4 \times b_4]) \tag{3}$$

(4) Division of interval rough numbers, "/",

$$IRN(A) / IRN(B) = ([a_1, a_2], [a_3, a_4]) / ([b_1, b_2], [b_3, b_4]) = ([a_1 / b_4, a_2 / b_3], [a_3 / b_2, a_4 / b_1]) \tag{4}$$

(5) Scalar multiplication of interval rough numbers where $k > 0$

$$k \times IRN(A) = k \times ([a_1, a_2], [a_3, a_4]) = ([k \times a_1, k \times a_2], [k \times a_3, k \times a_4]) \quad (5)$$

The intersection points of IRNs are obtained in the following way:

$$\mu_\alpha = \frac{RB(\alpha_{ui})}{RB(\alpha_{ui}) + RB(\alpha_{li})}; RB(\alpha_{ui}) = \alpha'^U - \alpha'^L; RB(\alpha_{li}) = \alpha^U - \alpha^L \quad (6)$$

$$\mu_\beta = \frac{RB(\beta_{ui})}{RB(\beta_{ui}) + RB(\beta_{li})}; RB(\beta_{ui}) = \beta'^U - \beta'^L; RB(\beta_{li}) = \beta^U - \beta^L \quad (7)$$

$$I(\alpha) = \mu_\alpha \cdot \alpha^L + (1 - \mu_\alpha) \cdot \alpha'^U \quad (8)$$

$$I(\beta) = \mu_\beta \cdot \beta^L + (1 - \mu_\beta) \cdot \beta'^U \quad (9)$$

### 3.2. A Novel Interval Rough SWARA Method

In this part of the paper, the extension of the SWARA method with interval rough numbers (IRN SWARA) is presented in detail [31]. IRN SWARA consists of the steps shown below:

Step 1: In this step, it is necessary to form a set of criteria on the basis of which the evaluation of variant solutions by some of the other methods should be performed.

Step 2: Creating a group of k decision makers (DMs) to determine the estimates of the significance of the criteria.

Step 3: Determining the $c_j$ matrix by identifying the most significant criterion and starting from that criterion, determining how much more significant the best $c_j$ criterion is compared to the $c_j$-$m$ criteria.

Step 4: Transformation of individual DMs' estimates into an interval group rough matrix IRNCj. Each individual DM's answer $k_1, k_2, ..., kn$ must be converted into a rough group matrix using equations.

$$IRN(C_j) = \left[c_j^L, c_j^U\right], \left[c_j'^L, c_j'^U\right]_{1 \times m} \quad (10)$$

Step 5: Calculating the normalized matrix $IRN(S_j)$:

$$IRN(S_j) = \left[s_j^L, s_j^U\right], \left[s_j'^L, s_j'^U\right] \quad (11)$$

The normalized matrix $IRN(S_j)$ is calculated by applying Equation (12):

$$IRN(S_j) = \frac{IRNC_j}{\max IRNC_j} \quad (12)$$

The first element of the $IRN(S_j)$ matrix, i.e., $\left[s_j^L, s_j^U\right], \left[s_j'^L, s_j'^U\right] = [1.00, 1.00], [1.00, 1.00]$, since $j = 1$. For other elements where $j > 1$, Equation (12) is calculated by applying Equation (13):

$$IRN(\overset{m}{\underset{j=2}{S_j}}) = \left[\frac{c_j^L}{\max c_j'^U}, \frac{c_j^U}{\max c_j'^L}\right], \left[\frac{c_j'^L}{\max c_j^U}, \frac{c_j'^U}{\max c_j^L}\right]_{1 \times m} \quad (13)$$

Step 6: Determining the $IRN(K_j)$ matrix:

$$IRN(K_j) = \left[k_j^L, k_j^U\right], \left[k_j'^L, k_j'^U\right]_{1 \times m} \quad (14)$$

by applying Equation (15):

$$IRN(\overset{m}{\underset{j=2}{K_j}}) = \left[s_j^L + 1, s_j^U + 1\right], \left[s_j'^L + 1, s_j'^U + 1\right]_{1 \times m} \quad (15)$$

Step 7: Calculation of recalculated weights—the $IRN(Q_j)$ matrix:

$$IRN(Q_j) = \left[q_j^L, q_j^U\right], \left[q_j'^L, q_j'^U\right]_{1 \times m} \tag{16}$$

The elements of the $IRN(Q_j)$ matrix are obtained as follows:

$$IRN(Q_j) \left[q_j^L = \left\{ \begin{array}{l} 1.00 j = 1 \\ \dfrac{q_{j-1}^L}{k_j'^U} j > 1 \end{array} \right. , q_j^U = \left\{ \begin{array}{l} 1.00 j = 1 \\ \dfrac{q_{j-1}^U}{k_j'^L} j > 1 \end{array} \right. \right], \left[q_j'^L = \left\{ \begin{array}{l} 1.00 j = 1 \\ \dfrac{q_{j-1}'^L}{k_j^U} j > 1 \end{array} \right. , q_j'^U = \left\{ \begin{array}{l} 1.00 j = 1 \\ \dfrac{q_{j-1}'^U}{k_j^L} j > 1 \end{array} \right. \right] \tag{17}$$

Step 8: Determining the matrix of relative weight values $IRN(W_j)$:

$$IRN(W_j) = \left[w_j^L, w_j^U\right], \left[w_j'^L, w_j'^U\right]_{1 \times m} \tag{18}$$

The individual weight values of the criteria are obtained by applying Equation (19):

$$\left[w_j^L, w_j^U\right], \left[w_j'^L, w_j'^U\right] = \left[ \frac{q_j^L}{\sum\limits_{j=1}^m q_j'^U}, \frac{q_j^U}{\sum\limits_{j=1}^m q_j'^L}, \frac{q_j'^L}{\sum\limits_{j=1}^m q_j^U}, \frac{q_j'^U}{\sum\limits_{j=1}^m q_j^L} \right] \tag{19}$$

### 3.3. A Novel Interval Rough ARAS Method

In this part of the paper, the extension of the ARAS method with interval rough numbers (IRN ARAS) is presented in detail [32]. IRN ARAS consists of the steps shown below:

Step 1. Defining the input parameters of the MCDM model. First, it is necessary to define a group of m criteria for the evaluation of n alternatives for a particular problem. Then, a team of k DMs is defined to evaluate the alternatives.

Step 2. Defining an interval group rough matrix. Here it is necessary to aggregate individual matrices into an interval group rough matrix, Equation (20). In the initial interval rough matrix, one row is added which implies the optimal values of the alternatives according to the criteria, depending on the type of criteria.

$$IRN(X_{ij}) = \begin{array}{c} \\ A_1 \\ A_2 \\ ... \\ A_m \end{array} \begin{array}{c} C_1 \qquad C_2 \qquad ... \qquad C_n \\ \left[ \begin{array}{cccc} IRN(x_{11}) & IRN(x_{12}) & ... & IRN(x_{1n}) \\ IRN(x_{21}) & IRN(x_{22}) & & IRN(x_{2n}) \\ ... & ... & ... & ... \\ IRN(x_{l1}) & IRN(x_{l2}) & ... & IRN(x_{ln}) \end{array} \right]_{m \times n} \end{array} \tag{20}$$

If the criterion is of the benefit type, the maximum values are included in the optimal values, while in the opposite case (cost type criterion), the minimum values are included.

Step 3. Determining the normalized interval rough matrix that is performed in a total of three phases depending on whether the criterion is of the benefit or cost type. In the first phase, normalization is performed if the criterion is of the benefit type (21).

$$(a)\ IRN(N_{ij}) = \frac{IRN(X_{ij})}{\sum\limits_{i=1}^m (X_{ij})} \Rightarrow n_{ij} = \left[ \frac{x_{ij}^L}{\sum\limits_{i=1}^m x_{ij}'^U}, \frac{x_{ij}^U}{\sum\limits_{i=1}^m x_{ij}'^L}, \frac{x_{ij}'^L}{\sum\limits_{i=1}^m x_{ij}^U}, \frac{x_{ij}'^U}{\sum\limits_{i=1}^m x_{ij}^L} \right] \ for\ c_1, c_2, c_3 \ldots c_n \in B \tag{21}$$

where $x_{ij}^L$ represents the lower and $x_{ij}^U$ the upper limit of the first part of the interval number (IRN), while $x_{ij}'^L$ represents the lower and $x_{ij}'^U$ the upper limit of the second part of the interval rough number.

In the second and third phase of normalization, the calculation is performed by applying Equations (22) and (23), respectively, when it comes to the cost criterion:

(b) for this type of criterion, Equation (22) is first applied to calculate the inverse values of the interval rough number $\overline{x_{ij}}$ for $\left[ x_{ij}{}^L, x_{ij}^U, x_{ij}{}'^L, x_{ij}'^U \right]$

$$\overline{x_{ij}} = \left[ \frac{1}{X_{ij}} \right] = \left[ \frac{1}{x_{ij}'^U}, \frac{1}{x_{ij}'^L}, \frac{1}{x_{ij}^U}, \frac{1}{x_{ij}^L} \right] \tag{22}$$

(c) and then Equation (23) is applied to complete the normalization process:

$$n_{ij} = \left[ \frac{\overline{x_{ij}^L}}{\sum\limits_{i=1}^{m} \overline{x_{ij}'^U}}, \frac{\overline{x_{ij}^U}}{\sum\limits_{i=1}^{m} \overline{x_{ij}'^L}}, \frac{\overline{x_{ij}'^L}}{\sum\limits_{i=1}^{m} \overline{x_{ij}^U}}, \frac{\overline{x_{ij}'^U}}{\sum\limits_{i=1}^{m} \overline{x_{ij}^L}} \right] \tag{23}$$

Step 4. Weighting the previous interval rough matrix using Equation (24):

$$IRN(W_n) = \left[ w_{ij}{}^L, w_{ij}^U, w_{ij}{}'^L, w_{ij}'^U \right]_{mxn} \Rightarrow \left. \begin{array}{l} w_{ij}^L = w_j^L \times n_{ij}^L \\ w_{ij}^U = w_j^U \times n_{ij}^U \\ w_{ij}'^L = w_j'^L \times n_{ij}'^L \\ w_{ij}'^U = w_j'^U \times n_{ij}'^U \end{array} \right\} i = O, 1, 2, \ldots, m \tag{24}$$

Step 5. Calculation of the matrix by summing the values from the previous matrix by rows (25).

$$IRN(S_i) = \left[ s_i^L, s_i^U, s_i'^L, s_i'^U \right] = \sum_{i=1}^{n} IRN(W_n) \Rightarrow \left. \begin{array}{l} s_i^L = \sum\limits_{i=1}^{n} w_i^L \\ s_i^U = \sum\limits_{i=1}^{n} w_i^U \\ s_i'^L = \sum\limits_{i=1}^{n} w_i'^L \\ s_i'^U = \sum\limits_{i=1}^{n} w_i'^U \end{array} \right\} \tag{25}$$

Step 6. Calculation of the utility function, Equation (26):

$$IRN(K_i) = \frac{IRN(S_i)}{IRN(S_o)} = \left[ \frac{s_i^L}{s_o'^U}, \frac{s_i^U}{s_o'^L}, \frac{s_i'^L}{s_o^U}, \frac{s_i'^U}{s_o^L} \right] \tag{26}$$

where $S_o$ denotes the value of the optimal alternative.

Step 7. Ranking alternatives in descending order.

## 4. Evaluation Strategies for Cleaner Production in Libyan Industry

The situation in Libyan industry is presented in more detail in Section 2, while this section of the paper presents an overall calculation for the evaluation of strategies for cleaner production based on the following eight criteria presented in Section 4.1:

### 4.1. MCDM Model Setting

For the purpose of determining the appropriate criteria to be used in the study, a preliminary list of these criteria has been established based on previous studies and was subsequently sent to a group of experts for evaluation and adjustment to suit the local industry environment. All of these experts have worked in the industrial sector for more than 20 years, whether in public or private industrial companies or in the Ministry of Industry as experts and consultants. After receiving their feedback, a list of the following criteria has been created:

C1: The extent to which the benefits of cleaner production are recognized. Perhaps one of the most significant obstacles to the implementation of any new policies or procedures is the lack of awareness and understanding of their benefits. With this new concept, the

barrier should be high. The last ten years in Libya and the instability it has experienced may have had a direct impact on the lack of human skills development in its institutions, and serious shortcomings are therefore expected regarding the understanding of these modern concepts, their importance, and their expected impact on institutions and the environment.

C2: How much senior management are interested in placing this concept at the top of their priorities. Many companies suffer from economic, financial, or technical problems, and they are struggling to overcome these problems, which is why they put these concepts at the bottom of their list of interests, especially with the unclear returns that can result from their application, or the lack of conviction that these returns are obtainable.

C3: The existence of databases on the current state of factories and companies. There is no reliable, up-to-date, and integrated database that can be consulted on the current situation of companies and plants. Indeed, it is difficult to even know the number of these plants, their categories, and their classifications. It is therefore difficult to put in place appropriate policies and for authorities in charge to take supportive measures related to these factories in the country.

C4: The availability of qualified human resources to implement these policies. The lack of qualified human resources to implement these policies remains one of the major obstacles. The educational system in Libya is also facing problems, as it falls at the bottom of the educational systems in the world. Moreover, companies and factories, in the absence of a clear vision, will not invest in training for these programs.

C5: The nature of processes used to apply financial incentives. The complex and bureaucratic financial system hinders the motivation of the different institutions to implement these programs.

C6: The modernization level of production technologies. The use of old production techniques or worn-out machinery does not constitute an incentive or support factor for the implementation of such policies.

C7: The level of the provision of project funding. It should be noted that failure to provide adequate funding hampers the conduct of technology replacement operations, as well as the qualifications of the human resources needed to implement these policies.

C8: The accessibility and, where applicable, the enforcement of laws and legislation regulating and stimulating such policies. These policies require a solid foundation of laws and legislation to support them, as well as to provide assistance in their implementation. Thus, the absence of such laws or failure in implementing them, if any, represents a serious obstacle.

All criteria are defined in such a way that they belong to a group of benefit criteria, which is important for the further development of the model, i.e., the evaluation of the following five strategies for cleaner production.

With the same methodology in which the evaluation criteria were identified, a set of appropriate strategies was also identified through the same group of experts from the industry sector. The list below illustrates such strategies:

S1: Launching awareness-raising campaigns to publicize these policies. These campaigns aim to increase management and personnel knowledge about the importance of these policies and their impact on the organization. Such campaigns are carried out in various ways such as workshops, seminars, meetings, social media posts, posters, and other methods.

S2: Putting in place more effective mechanisms to access information on the current situation of companies and factories. Before 2011, the Center of Industrial Information and Documentation belonging to the Ministry of Industry conducted an experiment by establishing an electronic database for all industrial enterprises in Libya, but it was subsequently discontinued. The adoption, updating, and development of such tools will still help to obtain the information needed easily and effectively.

S3: Developing financial and administrative systems that are efficient, encouraging, and attractive for investment. Developing decisions to encourage foreign financial investments such as the Minister of Economy Decree No. 207 of 2012. This edict was for es-

tablishing procedures for opening branches and representation offices of foreign companies, and identifying activities that foreign investors can practice through their companies.

S4: Developing policies and mechanisms for capacity building and skills. This is completed through the development of a general vision of the skills required for policy implementation, in addition to developing the mechanisms for implementing such policies as workshops, training programs, conferences, symposia, and other programs.

S5: Establishing legislation and laws to encourage these policies and activating the existing ones, as well as learning about the regulations and laws on clean production in the world and trying to update and activate the existing local laws.

### 4.2. Evaluation of the Criteria Using a Novel IRN SWARA Method

A total of five experts have participated in this research, expressing their preferences on the significance of the criteria, which are shown in Table 1. It is important to note that the evaluation of the significance of the criteria has been performed using interval numbers, which enables DMs to express their preferences more precisely.

**Table 1.** Evaluation of the criteria using interval numbers.

|  | E1 | E2 | E3 | E4 | E5 |
|---|---|---|---|---|---|
| **C1** | [3.5, 4] | [1, 1.5] | [2, 2] | [2, 2.5] | [2, 2] |
| **C2** | [3, 3] | [3, 3.5] | [1, 1] | [1.5, 2] | [1, 1] |
| **C3** | [6.5, 7] | [3, 3] | [5, 5.5] | [5, 5.5] | [5, 5] |
| **C4** | [1, 1.5] | [3.5, 4] | [2.5, 3] | [9, 9] | [1.5, 3] |
| **C5** | [9, 9] | [5, 5.5] | [7, 7.5] | [5, 6] | [8, 8] |
| **C6** | [2.5, 3] | [6, 6] | [2, 3] | [3, 4] | [3, 4] |
| **C7** | [4.5, 5] | [8, 8] | [9, 9] | [4, 4.5] | [6, 6] |
| **C8** | [5, 5] | [7, 8] | [4, 4.5] | [1, 2] | [5, 6] |

The experts who evaluated the criteria and strategies have the following experience:

Expert #01: Worked as a Consultant at the Ministry of Industry for more than twenty years, during which he evaluated several local industrial companies. He also served as a member of the Management Committee at three industrial companies. He has a Master's Degree in industrial engineering.

Expert #02: Worked as a Consultant at the Ministry of Industry for more than 10 years. He also served as General Manager/ Management Committee member at several industrial companies. He has a PhD in Industrial Engineering.

Expert #03: Served as a member of a Management Committee in many public and private industrial companies, and served as a Consultant at the Ministry of Industry for more than 10 years. He has a PhD in Industrial Engineering.

Expert #04: Worked at the Ministry of Industry for more than 25 years and carried out operations in designing the organizational structure for many industrial companies, as well as working in committees for evaluating several local companies. He has a Bachelor's degree in Management.

Expert #05: Worked with many industrial companies. He has an MSc degree in Industrial Engineering.

Since it is defined that IRN consists of two rough sequences, two classes of objects are identified. We take the example of the first criterion $w_1$ and $w'_1$: $w_1 = \{3.5, 1, 2, 2, 2\}$ and $w'_1 = \{4, 1.5, 2, 2.5, 2\}$. It is necessary to form rough sequences for each specified class of objects. For the stated first criterion, they are obtained for the 1th class of objects as follows:

$$Lim(1) = 1, \ \overline{Lim}(1) = \tfrac{1}{5}(3.5 + 1 + 2 + 2 + 2) = 2.1; \ RN(1) = [1, 2.1]$$

$$Lim(2) = \frac{1}{4}(1 + 2 + 2 + 2) = 1.75, \ \overline{Lim}(2) = \frac{1}{4}(3.5 + 2 + 2 + 2) = 2.38; \ RN(2) = [1.75, 2.38]$$

$$Lim(3.5) = \frac{1}{5}(3.5 + 1 + 2 + 2 + 2) = 2.1, \ \overline{Lim}(3.5) = 3.5; \ RN(3.5) = [2.1, 3.5]$$

For the 2nd class of objects, the following is obtained:

$Lim(1.5) = 1.5, \overline{Lim}(1.5) = \frac{1}{5}(4 + 1.5 + 2 + 2.5 + 2) = 2.4; RN(1.5) = [1.5, 2.4]$

$Lim(2) = \frac{1}{3}(1.5 + 2 + 2) = 1.83, \overline{Lim}(2) = \frac{1}{4}(4 + 2 + 2.5 + 2) = 2.63; RN(2) = [1.83, 2.63]$

$Lim(2.5) = \frac{1}{4}(1.5 + 2 + 2.5 + 2) = 2, \overline{Lim}(2.5) = \frac{1}{2}(4 + 2.5) = 3.25; RN(2.5) = [2, 3.25]$

$Lim(4) = \frac{1}{5}(4 + 1.5 + 2 + 2.5 + 2) = 2.4, \overline{Lim}(4) = 4; RN(4) = [2.4, 4]$

Based on the rough sequences, interval rough numbers are obtained:
$IRN(E1) = ([2.1, 3.5], [2.4, 4]), IRN(E2) = ([1, 2.1], [1.5, 2.4]), IRN(E3) = ([1.75, 2.38], [1.83, 2.63]), IRN(E4) = ([1.75, 2.38], [2, 3.25])$ and $IRN(E5) = ([1.75, 2.38], [1.83, 2.63])$.

Then, the final interval values that are an integral part of the IRNCj matrix are obtained in the following way (Table 2). Thus, the first four steps are: forming a set of criteria, creating a group of five DMs that express their preferences on the significance of the criteria, determining the $c_j$ matrix by determining the most significant criterion, and, starting from that criterion, determining how much more significant the best $c_j$ criterion is compared to $c_j$-$m$ criteria for each DM. The transformation of individual DMs' estimates into an interval group rough IRNCj matrix is also performed.

**Table 2.** Initial matrix IRNCj in rough SWARA.

| Criteria | Matrix $IRNC_j$ |
| :---: | :---: |
| C2 | [1.39, 2.46], [1.44, 2.76] |
| C1 | [1.67, 2.55], [1.91, 2.98] |
| C6 | [2.56, 4.19], [3.4, 4.67] |
| C4 | [1.91, 5.58], [2.7, 5.82] |
| C8 | [3.08, 5.6], [3.71, 6.46] |
| C3 | [4.28, 5.51], [4.34, 5.99] |
| C7 | [5, 7.67], [5.36, 7.73] |
| C5 | [5.74, 7.82], [6.31, 8.1] |

An example of obtaining a final interval number $IRN1 = [1.67, 2.55], [1.91, 2.98]$ is:

$$RN^L1 = \frac{1}{5}(2.1 + 1 + 1.75 + 1.75 + 1.75) = 1.67$$

$$RN^U1 = \frac{1}{5}(3.5 + 2.1 + 2.38 + 2.38 + 2.38) = 2.55$$

$$RN^L1' = \frac{1}{5}(2.4 + 1.5 + 1.83 + 2 + 1.83) = 1.91$$

$$RN^U1' = \frac{1}{5}(4 + 2.4 + 2.63 + 3.25 + 2.63) = 2.98$$

After that, normalization is performed in the fifth step of the IRN SWARA method using Equations (11)–(13). First, it is determined that an interval rough number represents the maximum individual values of all interval rough numbers from the *IRNCj* matrix. Table 3 shows the overall calculation of the weight values of the criteria using the IRN SWARA method.

$$\text{max} IRNC_j = [5.74, 7.82], [6.31, 8.10]$$

The first element of the $IRN(S_j)$ matrix is $\left[s_j^L, s_j^U\right], \left[s_j'^L, s_j'^U\right] = [1.00, 1.00], [1.00, 1.00]$, since $j = 1$. For other elements $j > 1$, Equation (13) is applied, e.g.,:

$$IRN(S_1) = \left[\frac{1.67}{8.10}, \frac{2.55}{6.31}\right], \left[\frac{1.91}{7.82}, \frac{2.98}{5.74}\right] = [0.21, 0.40], [0.24, 0.52].$$

**Table 3.** Overview of the calculation of criterion weights through the IRN SWARA steps.

|  | $IRN\left(S_j\right)$ | $IRN\left(K_j\right)$ | $IRN\left(Q_j\right)$ | $IRN\left(W_j\right)$ |
|---|---|---|---|---|
| C2 | [1.00, 1.00], [1.00, 1.00] | [1.00, 1.00], [1.00, 1.00] | [1.00, 1.00], [1.00, 1.00] | [0.26, 0.39], [0.29, 0.43] |
| C1 | [0.21, 0.4], [0.24, 0.52] | [1.21, 1.4], [1.24, 1.52] | [0.66, 0.8], [0.71, 0.83] | [0.17, 0.31], [0.21, 0.35] |
| C6 | [0.32, 0.66], [0.43, 0.81] | [1.32, 1.66], [1.43, 1.81] | [0.36, 0.56], [0.43, 0.63] | [0.1, 0.22], [0.13, 0.27] |
| C4 | [0.24, 0.88], [0.35, 1.01] | [1.24, 1.88], [1.35, 2.01] | [0.18, 0.42], [0.23, 0.51] | [0.05, 0.16], [0.07, 0.22] |
| C8 | [0.38, 0.89], [0.47, 1.13] | [1.38, 1.89], [1.47, 2.13] | [0.08, 0.28], [0.12, 0.37] | [0.02, 0.11], [0.04, 0.16] |
| C3 | [0.53, 0.87], [0.55, 1.04] | [1.53, 1.87], [1.55, 2.04] | [0.04, 0.18], [0.06, 0.24] | [0.01, 0.07], [0.02, 0.1] |
| C7 | [0.62, 1.22], [0.69, 1.35] | [1.62, 2.22], [1.69, 2.35] | [0.02, 0.11], [0.03, 0.15] | [0, 0.04], [0.01, 0.06] |
| C5 | [0.71, 1.24], [0.81, 1.41] | [1.71, 2.24], [1.81, 2.41] | [0.01, 0.06], [0.01, 0.09] | [0, 0.02], [0, 0.04] |

The elements of the $IRN(K_j)$ matrix are obtained by applying Equation (15) and adding the values of 1.000 to all values of the interval rough number, except for the first criterion in terms of significance (C2) whose $IRN(K_2) = [1.00, 1.00], [1.00, 1.00]$.

The calculation of recalculated weights for the $IRN(Q_j)$ matrix is performed in the seventh step using Equation (17) as follows:

$$IRN(Q_1)\left[q_1^L = \frac{q_2^L}{k_1'^U} = \frac{1.000}{1.519} = 0.658, q_1^U = \frac{q_2^U}{k_1'^L} = \frac{1.000}{1.244} = 0.804\right],$$

$$\left[q_1'^L = \frac{q_2'^L}{k_1^U} = \frac{1.000}{1.404} = 0.712, q_1'^U = \frac{q_2'^U}{k_1^L} = \frac{1.000}{1.206} = 0.829\right]$$

In the eighth step, the matrix of relative weight values $IRN(W_j)$ is determined using Equation (19) as follows:

$$\left[w_2^L, w_2^U\right], \left[w_2'^L, w_2'^U\right] = \left[\frac{q_2^L}{\sum\limits_{j=1}^{m} q_j'^U}, \frac{q_2^U}{\sum\limits_{j=1}^{m} q_j'^L}, \frac{q_2'^L}{\sum\limits_{j=1}^{m} q_j^U}, \frac{q_2'^U}{\sum\limits_{j=1}^{m} q_j^L}\right] = \left[\frac{1.000}{3.817}, \frac{1.000}{2.594}, \frac{1.000}{3.412}, \frac{1.000}{2.353}\right]$$

Table 3 shows the rank of obstacles to the application of the concept of cleaner production according to their importance. The table shows that the most important obstacle is that companies and factories do not prioritize this concept. This can be explained by the fact that the government's industrial sector is mostly underproductive or underdeveloped, with exceptions such as the Libyan Iron and Steel Company, one of the largest domestic industrial companies. In the private sector, the industrial experience is somewhat new. Most of the private industrial companies in Libya are less than 20 years old, and these companies mostly started as a private family activity and then grew up. These companies struggle to remain in the market under the competition of imported goods, both in terms of price and quality. Although the concept of cleaner production can improve the performance of these companies, they do not have sufficient experience, as most companies (especially in the private sector) lack research and development offices and training programs. The second obstacle is the lack of awareness of the importance of these strategies and the inability to recognize the consequences of environmental problems. Low awareness of environmental issues is a common problem in the country, causing many environmental problems.

*4.3. Evaluation of Cleaner Production Strategies Using a Novel IRN ARAS Method*

In order to obtain an initial interval rough matrix, cleaner production strategies are evaluated by five experts using interval numbers as shown in Table 4.

**Table 4.** Evaluation of cleaner production strategies by five DMs using interval numbers.

| | S1 | | | | | S2 | | | | | S3 | | | | | S4 | | | | | S5 | | | | |
| | E1 | E2 | E3 | E4 | E5 | E1 | E2 | E3 | E4 | E5 | E1 | E2 | E3 | E4 | E5 | E1 | E2 | E3 | E4 | E5 | E1 | E2 | E3 | E4 | E5 |
|---|---|---|---|---|---|---|---|---|---|---|---|---|---|---|---|---|---|---|---|---|---|---|---|---|---|
| C1 | (9,9) | (9,9) | (8,8) | (5,6) | (8,8) | (7,7) | (3,4) | (4,4) | (4,5) | (4,5) | (6,7) | (6,6) | (6,7) | (5,6) | (7,7) | (1,2) | (3,3) | (2,3) | (6,7) | (6,6) | (5,6) | (7,7) | (5,5) | (5,5) | (2,3) |
| C2 | (8,8) | (8,9) | (5,6) | (3,3) | (8,9) | (4,5) | (6,6) | (2,3) | (2,3) | (5,5) | (5,6) | (5,5) | (2,3) | (1,2) | (6,6) | (7,7) | (6,7) | (6,7) | (5,5) | (4,5) | (6,6) | (6,7) | (3,3) | (3,4) | (4,4) |
| C3 | (2,3) | (1,2) | (1,1) | (1,2) | (4,4) | (9,9) | (9,9) | (6,7) | (2,3) | (6,6) | (1,1) | (1,2) | (3,3) | (2,3) | (4,5) | (7,7) | (6,7) | (6,6) | (5,5) | (2,3) | (4,5) | (1,2) | (3,4) | (2,3) | (4,5) |
| C4 | (3,4) | (1,1) | (5,6) | (5,6) | (6,7) | (1,1) | (1,2) | (2,3) | (7,7) | (5,6) | (3,4) | (4,4) | (6,7) | (6,7) | (5,5) | (7,7) | (6,6) | (6,7) | (9,9) | (9,9) | (4,5) | (4,5) | (4,4) | (8,8) | (4,4) |
| C5 | (3,3) | (3,3) | (3,3) | (3,3) | (3,4) | (1,2) | (4,5) | (2,3) | (3,3) | (2,3) | (9,9) | (9,9) | (7,8) | (2,3) | (5,6) | (7,8) | (7,7) | (6,6) | (4,5) | (1,2) | (8,8) | (9,9) | (5,6) | (4,5) | (3,4) |
| C6 | (1,2) | (1,2) | (3,4) | (5,5) | (7,7) | (5,6) | (2,3) | (4,5) | (5,6) | (5,5) | (9,9) | (7,7) | (6,7) | (6,7) | (3,4) | (6,7) | (6,6) | (7,8) | (2,3) | (1,2) | (7,8) | (3,4) | (3,4) | (6,7) | (3,3) |
| C7 | (2,3) | (4,5) | (3,3) | (3,4) | (3,4) | (5,5) | (1,2) | (1,2) | (2,3) | (2,3) | (8,8) | (9,9) | (7,7) | (2,3) | (3,4) | (7,7) | (6,7) | (5,6) | (2,3) | (5,5) | (8,9) | (8,8) | (7,7) | (4,4) | (3,4) |
| C8 | (5,5) | (3,4) | (6,7) | (9,9) | (6,7) | (3,3) | (1,2) | (4,5) | (9,9) | (5,6) | (6,7) | (9,9) | (8,8) | (8,8) | (4,5) | (7,8) | (7,7) | (6,6) | (6,7) | (8,8) | (9,9) | (9,9) | (8,8) | (9,9) | (6,7) |

In the same way as presented in the previous section of the paper, the individual interval estimates are transformed into an interval rough matrix, which is the initial matrix for evaluating strategies for cleaner production using the IRN ARAS method. The initial interval rough matrix is shown in Table 5.

**Table 5.** Initial interval rough matrix for the IRN ARAS method.

|     | C1 | C2 | C3 | C4 | C5 | C6 | C7 | C8 |
|-----|-----|-----|-----|-----|-----|-----|-----|-----|
| S1 | [6.92, 8.56] [7.33, 8.60] | [5.24, 7.53] [5.43, 8.33] | [1.21, 2.48] [1.75, 3.08] | [2.8, 5.08] [3.36, 6.04] | [2.36, 2.84] [2.36, 3.25] | [1.91, 4.96] [2.78, 5.27] | [2.65, 3.35] [3.36, 4.25] | [4.56, 7.06] [5.28, 7.55] |
| S2 | [3.73, 5.13] [4.4, 5.67] | [2.56, 4.99] [3.68, 5.09] | [4.83, 7.88] [5.29, 8.18] | [1.76, 4.81] [2.26, 5.43] | [1.75, 3.08] [2.69, 3.74] | [3.52, 4.79] [4.33, 5.6] | [1.44, 3.08] [2.4, 3.67] | [2.66, 6.33] [3.37, 6.78] |
| S3 | [5.65, 6.35] [6.36, 6.84] | [5.5, 6.5] [3.33, 5.41] | [1.46, 2.98] [1.96, 3.68] | [4.02, 5.54] [4.63, 6.23] | [4.59, 8.05] [5.43, 8.33] | [4.94, 7.44] [5.91, 7.66] | [3.86, 7.61] [4.57, 7.74] | [5.8, 8.08] [6.57, 8.5] |
| S4 | [5.36, 7.48] [5.96, 7.68] | [2.92, 4.56] [3.52, 4.79] | [1.79, 3.92] [2.18, 4.26] | [6.33, 8.41] [6.77, 8.37] | [1.91, 3.32] [2.63, 4.23] | [2.29, 4.87] [3.29, 5.87] | [2.92, 4.25] [3.94, 4.83] | [5.08, 7.24] [5.57, 7.5] |
| S5 | [3.91, 5.66] [4.32, 6.04] | [3.63, 5.23] [3.88, 5.76] | [2.02, 3.54] [3.02, 4.54] | [4.16, 5.44] [4.44, 6.08] | [4.26, 7.43] [5.13, 7.71] | [3.43, 5.34] [4.01, 6.44] | [4.63, 7.28] [5.03, 7.66] | [7.52, 8.79] [7.94, 8.83] |
| So | [6.92, 8.56] [7.33, 8.60] | [5.5, 7.53] [5.43, 8.33] | [4.83, 7.88] [5.29, 8.18] | [6.33, 8.41] [6.77, 8.37] | [4.59, 8.05] [5.43, 8.33] | [4.94, 7.44] [5.91, 7.66] | [4.63, 7.61] [5.03, 7.74] | [7.52, 8.79] [7.94, 8.83] |

In the third step of the IRN ARAS method, normalization (Table 6) is performed through three steps if there are criteria that belong to both groups. However, since all the criteria are of the benefit type in this paper, only the first phase of the third step, i.e., Equation (21), is applied as follows:

$$n_{11} = \left[ \frac{x_{11}^L}{\sum\limits_{i=1}^{m} x_{ij}^{\prime U}}, \frac{x_{11}^U}{\sum\limits_{i=1}^{m} x_{ij}^{\prime L}}, \frac{x_{11}^{\prime L}}{\sum\limits_{i=1}^{m} x_{ij}^{U}}, \frac{x_{11}^{\prime U}}{\sum\limits_{i=1}^{m} x_{ij}^{L}} \right] = \left[ \frac{6.92}{43.43}, \frac{8.56}{35.70}, \frac{7.33}{41.74}, \frac{8.60}{32.49} \right] = [0.16, 0.24], [0.18, 0.26]$$

**Table 6.** Normalized interval rough matrix for the IRN ARAS method.

|     | C1 | C2 | C3 | C4 | C5 | C6 | C7 | C8 |
|-----|-----|-----|-----|-----|-----|-----|-----|-----|
| S1 | [0.16, 0.24] [0.18, 0.26] | [0.14, 0.3] [0.15, 0.33] | [0.04, 0.13] [0.06, 0.19] | [0.07, 0.18] [0.09, 0.24] | [0.07, 0.12] [0.07, 0.17] | [0.05, 0.19] [0.08, 0.25] | [0.07, 0.14] [0.1, 0.21] | [0.1, 0.19] [0.11, 0.23] |
| S2 | [0.09, 0.14] [0.11, 0.17] | [0.07, 0.2] [0.1, 0.2] | [0.15, 0.4] [0.18, 0.51] | [0.04, 0.17] [0.06, 0.21] | [0.05, 0.13] [0.08, 0.19] | [0.09, 0.18] [0.12, 0.27] | [0.04, 0.13] [0.07, 0.18] | [0.06, 0.17] [0.07, 0.2] |
| S3 | [0.13, 0.18] [0.15, 0.21] | [0.15, 0.26] [0.09, 0.21] | [0.05, 0.15] [0.07, 0.23] | [0.1, 0.2] [0.12, 0.25] | [0.13, 0.34] [0.17, 0.43] | [0.13, 0.28] [0.17, 0.36] | [0.11, 0.31] [0.14, 0.38] | [0.12, 0.22] [0.14, 0.26] |
| S4 | [0.12, 0.21] [0.14, 0.24] | [0.08, 0.18] [0.1, 0.19] | [0.06, 0.2] [0.08, 0.26] | [0.16, 0.3] [0.18, 0.33] | [0.05, 0.14] [0.08, 0.22] | [0.06, 0.19] [0.09, 0.28] | [0.08, 0.17] [0.12, 0.24] | [0.11, 0.2] [0.12, 0.23] |
| S5 | [0.09, 0.16] [0.1, 0.19] | [0.1, 0.21] [0.11, 0.23] | [0.06, 0.18] [0.11, 0.28] | [0.1, 0.19] [0.12, 0.24] | [0.12, 0.31] [0.16, 0.4] | [0.09, 0.2] [0.12, 0.31] | [0.13, 0.3] [0.15, 0.38] | [0.16, 0.24] [0.17, 0.27] |
| So | [0.16, 0.24] [0.18, 0.26] | [0.15, 0.30] [0.15, 0.33] | [0.15, 0.40] [0.18, 0.51] | [0.16, 0.3] [0.18, 0.33] | [0.16, 0.3] [0.18, 0.33] | [0.13, 0.28] [0.17, 0.36] | [0.13, 0.31] [0.15, 0.38] | [0.16, 0.24] [0.17, 0.27] |

After that, in the fourth step, the previous normalized interval rough matrix is weighted with the criterion weights obtained by applying the IRN SWARA method. The weighted normalized interval rough matrix is shown in Table 7.

The weighted normalized interval rough matrix is obtained by applying Equation (24) as follows:

$$\left. \begin{array}{l} w_{11}^L = w_1^L \times n_{11}^L \\ w_{11}^U = w_1^U \times n_{11}^U \\ w_{11}^{\prime L} = w_1^{\prime L} \times n_{11}^{\prime L} \\ w_{i11}^{\prime U} = w_1^{\prime U} \times n_{11}^{\prime U} \end{array} \right\} = \left. \begin{array}{l} 0.03 = 0.16 \times 0.17 \\ 0.07 = 0.24 \times 0.31 \\ 0.04 = 0.18 \times 0.21 \\ 0.09 = 0.26 \times 0.35 \end{array} \right\} = [0.03, 0.07], [0.04, 0.09]$$

**Table 7.** Weighted normalized interval rough matrix for the IRN ARAS method.

|    | C1 | C2 | C3 | C4 | C5 | C6 | C7 | C8 |
|----|----|----|----|----|----|----|----|----|
| S1 | [0.03, 0.07] | [0.04, 0.11] | [0, 0.01] | [0, 0.03] | [0, 0] | [0, 0.04] | [0, 0.01] | [0, 0.02] |
|    | [0.04, 0.09] | [0.04, 0.14] | [0, 0.02] | [0.01, 0.05] | [0, 0.01] | [0.01, 0.07] | [0, 0.01] | [0, 0.04] |
| S2 | [0.01, 0.04] | [0.02, 0.08] | [0, 0.03] | [0, 0.03] | [0, 0] | [0.01, 0.04] | [0, 0.01] | [0, 0.02] |
|    | [0.02, 0.06] | [0.03, 0.09] | [0, 0.05] | [0, 0.05] | [0, 0.01] | [0.02, 0.07] | [0, 0.01] | [0, 0.03] |
| S3 | [0.02, 0.06] | [0.04, 0.1] | [0, 0.01] | [0, 0.03] | [0, 0.01] | [0.01, 0.06] | [0, 0.01] | [0, 0.02] |
|    | [0.03, 0.07] | [0.03, 0.09] | [0, 0.02] | [0.01, 0.05] | [0, 0.02] | [0.02, 0.1] | [0, 0.02] | [0.01, 0.04] |
| S4 | [0.02, 0.06] | [0.02, 0.07] | [0, 0.01] | [0.01, 0.05] | [0, 0] | [0.01, 0.04] | [0, 0.01] | [0, 0.02] |
|    | [0.03, 0.08] | [0.03, 0.08] | [0, 0.03] | [0.01, 0.07] | [0, 0.01] | [0.01, 0.07] | [0, 0.02] | [0, 0.04] |
| S5 | [0.02, 0.05] | [0.03, 0.08] | [0, 0.01] | [0, 0.03] | [0, 0.01] | [0.01, 0.04] | [0, 0.01] | [0, 0.03] |
|    | [0.02, 0.07] | [0.03, 0.1] | [0, 0.03] | [0.01, 0.05] | [0, 0.01] | [0.01, 0.08] | [0, 0.02] | [0.01, 0.04] |
| So | [0.03, 0.07] | [0.04, 0.11] | [0, 0.03] | [0.01, 0.05] | [0, 0.01] | [0.01, 0.06] | [0, 0.01] | [0, 0.03] |
|    | [0.04, 0.09] | [0.04, 0.14] | [0, 0.05] | [0.01, 0.07] | [0, 0.02] | [0.02, 0.1] | [0, 0.02] | [0.01, 0.04] |

Table 8 shows the final results of the applied model, with details of the last three steps from the IRN ARAS method.

**Table 8.** Final results and ranking of strategies for cleaner production.

|    | $IRN(S_i)$ | $IRN(K_i)$ | Rank |
|----|-----------|-----------|------|
| S1 | [0.07, 0.3], [0.1, 0.43] | [0.14, 2.38], [0.28, 4.67] | 1 |
| S2 | [0.05, 0.24], [0.08, 0.37] | [0.09, 1.94], [0.21, 4.03] | 5 |
| S3 | [0.08, 0.3], [0.1, 0.42] | [0.15, 2.42], [0.26, 4.6] | 2 |
| S4 | [0.06, 0.27], [0.09, 0.4] | [0.11, 2.15], [0.24, 4.34] | 4 |
| S5 | [0.06, 0.26], [0.09, 0.41] | [0.11, 2.1], [0.23, 4.44] | 3 |
| So | [0.09, 0.37], [0.13, 0.54] | | |

In the fifth step, the $IRN(S_i)$ matrix is calculated by summing the values by rows from the previous weighted matrix. The elements of the $IRN(S_i)$ matrix are obtained using Equation (25), e.g.,:

$$S_1 = [0.07, 0.30, 0.10, 0.43] = \left.\begin{array}{l} s_1^L = \sum_{i=1}^{n} w_1^L \\ s_1^U = \sum_{i=1}^{n} w_1^U \\ s_1'^L = \sum_{i=1}^{n} w_1'^L \\ s_1'^U = \sum_{i=1}^{n} w_1'^U \end{array}\right\}$$

Then, in the sixth step, the utility function is calculated using Equation (26), e.g.,:

$$K_1 = \frac{S_1}{S_o} = \left[\frac{0.07}{0.536}, \frac{0.30}{0.13}\right], \left[\frac{0.10}{0.37}, \frac{0.43}{0.091}\right] = [0.14, 2.38], [0.28, 4.67]$$

As the last step in the IRN ARAS method, the ranking is performed, which is also shown in Table 8.

Table 8 shows the rankings of appropriate strategies to overcome obstacles to the application of cleaner production concepts in the Libyan industrial sector. At the forefront of these strategies, we find awareness-raising campaigns to introduce cleaner production concepts, which will help companies in the sector, whether at the level of senior management, owners, or employees, to recognize the importance of these concepts and their direct impact on their performance. It will undoubtedly contribute to the conviction of these companies in the importance of applying such concepts, and thus in seeking to implement it. In the second place, there is the development of effective, encouraging, and attractive financial and management systems for foreign investment, which will certainly

contribute to replacement operations, factory development, and the use of new technologies, thereby stimulating the application of this concept. The third strategy is the need to update legislation and laws on cleaner production. Such laws, in turn, will raise awareness of environmental issues and limit environmental damage.

## 5. Sensitivity Analysis and Discussion of Results

The comparison of the IRN ARAS is based on a set of characteristics that MCDM models need to satisfy in order to adequately model the decision-making process presented in the previous section. The following factors are considered: checking the robustness of the solution compared to other MCDM models; adequacy in adapting to changes in the criteria; adequacy for supporting group decision making; the number of alternatives and criteria; and modeling of uncertainty.

### 5.1. Checking the Robustness of the Solution Compared to Other MCDM Models

In this part, the verification of the above proposed model through several phases is presented. First, a comparative analysis (CA) of the obtained results is performed with the already developed methods: the Interval Rough SAW [33], Interval Rough CoCoSo [34], and Interval Rough COPRAS [35] methods. The results of the CA are shown in Figure 2, while the ranks in the CA are shown in Figure 3.

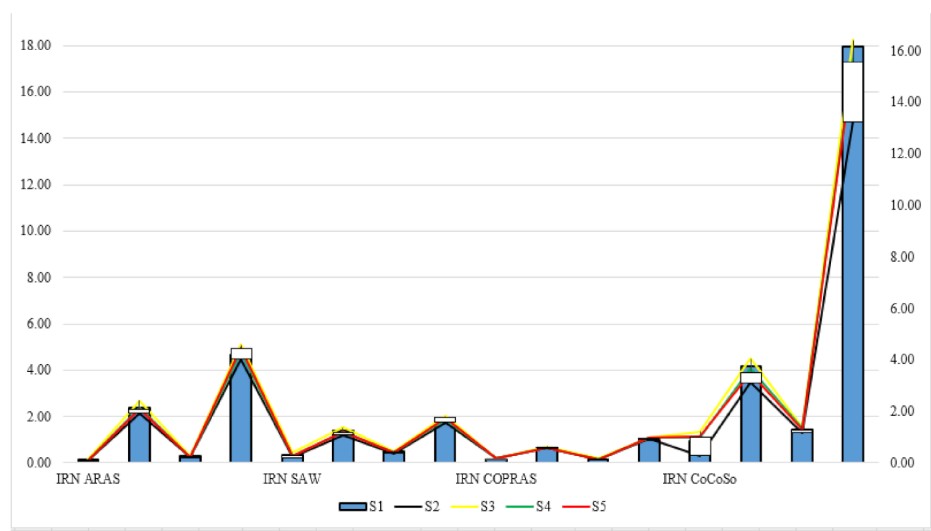

**Figure 2.** Results of comparative analysis.

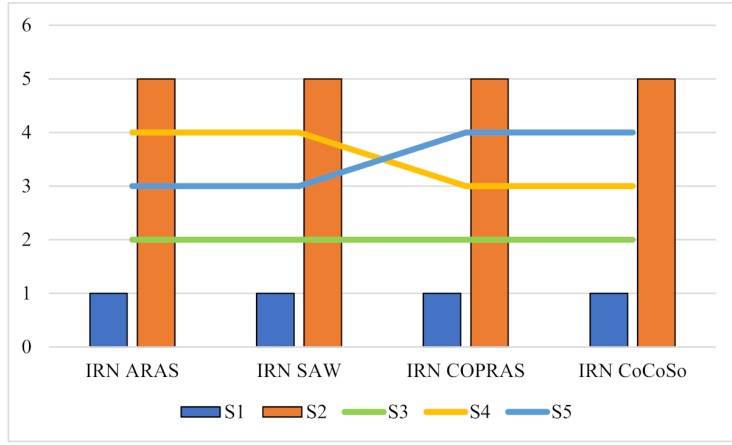

**Figure 3.** Ranks in a comparative analysis.

Figure 2 shows the values of IRNs in a CA. IRN COPRAS have the smallest range, while the interval values in IRN CoCoSo have the largest range (0.93–17.95), and the IRNs and IRN SAW have a smaller range (0.358–1.882). In general, the values of IRNs and their ranges vary depending on the applied methods, but the results using the integrated IRN SWARA–RN ARAS model have been confirmed (Figure 3).

Figure 3 shows that in a CA, there is a change in the ranks of the IRN COPRAS and IRN CoCoSo methods when it comes to the S4 and S5 strategies. Applying these two methods, the S4 and S5 strategies change their positions. The S4 Strategy has obtained better results compared to IRN ARAS and IRN SAW, so it is in the third position, while S5 has fallen to the fourth place. As for other strategies, there are no changes, so it can be concluded that the model in a comparative analysis is slightly sensitive, but stable when it comes to the best-placed strategies.

### 5.2. Adequacy for Changes in Criteria

In the second phase, the influence of changes in the value of the weight coefficients of the criteria on the ranking results are analyzed. The changes in the value of the weight coefficients are performed through 99 scenarios simulating the change in the value of the weight coefficient for the most influential criterion (C2). In the first scenario, the value of criterion C2 is reduced by 1%, while the values of the remaining criteria are proportionally corrected by applying Expression (27):

$$w_n = \frac{1 - w_1}{1 - w_1^*} \cdot w_n^* \tag{27}$$

where $w_1^*$ is the corrected value of the criterion weight C2, $w_n^*$ is the reduced value of the considered criterion, $w_n$ is the original value of the considered criterion, and $w_1$ is the original value of criterion C2, which is used for the calculation of the initial rank. The changes in the values of the score functions in the alternatives are shown in Figure 4.

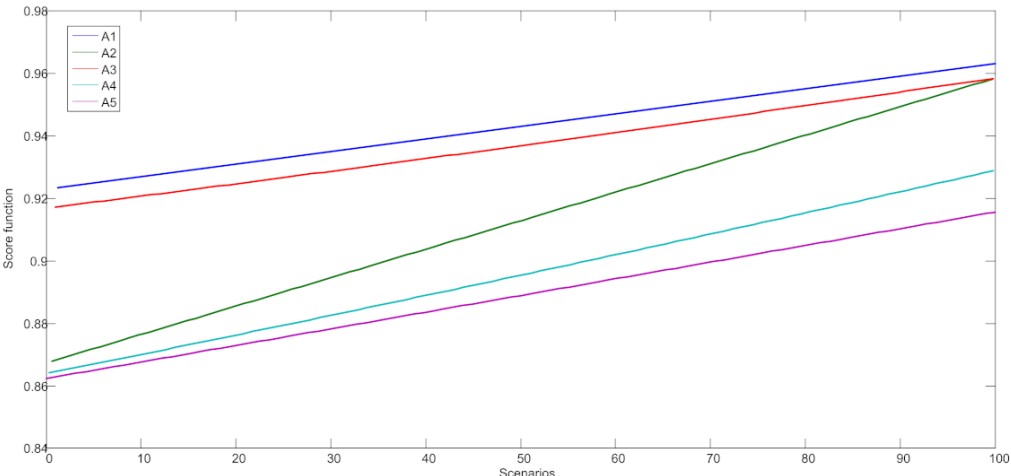

**Figure 4.** Influence of changes in criterion weight coefficients on changes in the score functions of the alternatives.

The results show that there is a linear dependence of changes in the score function on changes in the weight coefficients of the criteria. The linear dependence is a consequence of the nature of Equation (24), which is used to introduce the weights of the criteria into the initial decision matrix. Based on the results shown in Figure 3, we can conclude that changes in the values of the weight coefficients of the criteria affect the changes in the values of the score functions of the alternatives. We can also conclude that these changes do not lead to changes in the ranking of the alternatives, which indicates the fact that the initial rank is confirmed and credible.

### 5.3. Adequacy to Supporting Group Decision Making

IRN ARAS as well as the classical ARAS method allow the aggregation of expert preferences in the case of group decision making processes. In the case of the classical ARAS method, it is necessary to define special expressions for the fusion of expert estimates/preferences. In some studies, the authors propose aggregation of values using arithmetic averaging [36] or geometric averaging. However, simple aggregation of group decision making eliminates the uncertainties and inaccuracies that exist when presenting expert preferences in an initiated decision matrix. On the other hand, the IRN ARAS method applies an interval methodology based on the application of IRNs. These numbers allow for the presentation of expert preferences while respecting the ambiguities and inaccuracies that exist in a decision-making process. The fusion of IRNs objectifies the decision-making process. Therefore, although both methods can be applied to group decision making, the advantage is in favor of the IRN ARAS method, due to its more comprehensive mathematical algorithm that takes into account uncertainties and inaccuracies when aggregating expert preferences.

### 5.4. The Number of Alternatives and Criteria

Both the IRN ARAS method and the classical ARAS method do not impose restrictions when it comes to the number of alternatives or criteria used in the decision-making process. On the other hand, there are certain real situations in which there are uncertainties and a lack of information on certain alternatives. In such situations, the advantages of the IRN ARAS model are emphasized, enabling the recognition of uncertainties and the lack of information on alternatives. Therefore, the selection of a method depends on the conditions under which the decision is made.

### 5.5. Modeling of Uncertainty

The IRN ARAS method uses interval rough numbers to model uncertainty, while the classical ARAS methodology uses crisp values. The application of crisp values significantly simplifies the mathematical formulation of the considered model. On the other hand, the introduction of rough numbers in the ARAS method complicates its mathematical formulation. The IRN ARAS method implies the estimation of uncertainties and inaccuracies using rough numbers, which increases the mathematical complexity of the IRN ARAS method compared to the classical ARAS method.

## 6. Conclusions

In this research, a multi-criteria model of evaluation of five strategies for cleaner production based on eight sustainable criteria in Libyan industry has been considered. For this purpose, a novel integrated IRN SWARA–IRN ARAS model has been developed, the advantages of which can be seen through the following. The first advantage involves the relatively small number of steps (especially after transformation into an interval rough matrix) for both methods. When it comes to IRN SWARA, at the very beginning, it defines the most significant criterion and compares it with others, taking into account decision-makers' dilemmas, which are very common. When it comes to IRN ARAS, it is defined an optimal variant solution, which serves as an opportunity for more precise decision making. Then, uncertainties, dilemmas, and subjectivity in the decision-making process are reduced by applying interval rough numbers instead of crisp numbers.

Here, we can identify four main contributions and novelties of this study:

(1) The presentation of a novel IRN SWARA-ARAS model that allows for an objective evaluation of strategies of cleaner production in an uncertain environment;

(2) An improved MCDM methodology has been proposed, which is a powerful management tool for decision making;

(3) The presented methodology enables the evaluation of alternatives despite the uncertainties in the decision-making process and the lack of quantitative information;

(4)    The IRN SWARA–ARAS model enables a flexible decision-making process and serves as a useful reference for researchers in the field of cleaner production and other operational areas.

The main limitation of the IRN SWARA–ARAS model is its complex mathematical algorithm for calculating interval rough values. The increase in the number of experts in the research further complicates the application of this algorithm. This limitation can be eliminated by creating a software solution that will be user-oriented and that will allow a wider application of the presented approach in solving real world problems. During the work on this study, the authors developed a software solution based on the application of MATLAB and Microsoft Excel software packages. The software solution was used to validate the IRN SWARA–ARAS model solution.

Further research is recommended on a larger sample of respondents and considering the possibility of increasing the number of evaluation criteria through the introduction of sub-criteria within the groups of criteria. Further research should also be directed towards the development of a universal decision-making tool based on the IRN SWARA–ARAS methodology that allows the application of a different number of criteria/sub-criteria for decision making. In addition, this study may open new directions for future research in terms of methodological [37–39] and practical applications. It is recommended to focus future research on extending the proposed methodology by using hybrid fuzzy-rough numbers. This extension will allow the exploitation of the advantages of fuzzy theory and rough theory at the same time and the formation of limit values of interval fuzzy-rough numbers based on uncertainties that exist in the preferences of experts. This would eliminate the introduction of assumptions when defining the limit values of the interval of classical fuzzy numbers.

**Author Contributions:** Conceptualization I.B., Ž.S. and I.T.; methodology, Ž.S., D.P. and E.K.Z.; validation, I.T. and R.B.; formal analysis, I.T., Ž.S. and D.P.; investigation, I.B.; writing—original draft preparation, I.B., R.B. and Ž.S.; writing—review and editing, E.K.Z. and D.P., project administration E.K.Z. and R.B. All authors have read and agreed to the published version of the manuscript.

**Funding:** This research received no external funding.

**Institutional Review Board Statement:** Not applicable.

**Informed Consent Statement:** Not applicable.

**Data Availability Statement:** The data used to support the findings of this study are included within this article. However, the reader may contact the corresponding author for more details on the data.

**Conflicts of Interest:** The authors declare no conflict of interest.

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
