# Peer review of "A Novel Hybrid Interval Rough SWARA–Interval Rough ARAS Model for Evaluation Strategies of Cleaner Production"

_sustainability, doi:10.3390/su14074343_

Round 1
Reviewer 1 Report
The work is interested. The following are the suggestions.
- Add the objective of the work in the introduction section.
- There is a need to update the literature with some more articles such as A novel approach for solving rough multi - objective transportation problem: Development and Prospects; Cq-ROFRS: Covering q-rung orthopair fuzzy rough sets and its applications to multi-attribute decision-making process; Study on multi-objective nonlinear-programming problem with rough parameters,;
- Check all equations and symbols. Some are undefined.
- Describe the importance of rough features in the decision-making process.
- Add advantages of SWARA and ARAS method over the existing studies.
- Discussion of the results needs to be added in details.
- Conclusion section should be rewritten by adding the major finding, advantages and future scope of the work.
- Correct the reference style as per the journal format. It should be uniform throughout the study.
- Future research work direction should be added with articles: On the Application of a Lexicographic Method to Fuzzy Linear Programming Problems; Similarity-Distance Decision-Making Technique and its Applications via Intuitionistic Fuzzy Pairs
Author Response
Reviewer 1:
Thank you very much for the useful suggestions. We accepted all of the suggestions and we are sure that this will improve the quality and contribute to a better understanding of the paper.
The work is interested. The following are the suggestions.
----------------------------------------------------------------------------------------
Comment 1: Add the objective of the work in the introduction section.
Reply: Objectives in introduction are as follow:
The most important objectives that can be achieved by the research are the following:
1) to provide the best possible basis for the application of cleaner production in Libyan industry,
2) development of a novel integrated interval rough SWARA – rough ARAS interval taking into account uncertainties in decision making, and
3) additional enrichment of rough set theory and MCDM areas through the development and application of this integrated model.
Comment 2: There is a need to update the literature with some more articles such as A novel approach for solving rough multi - objective transportation problem: Development and Prospects; Cq-ROFRS: Covering q-rung orthopair fuzzy rough sets and its applications to multi-attribute decision-making process; Study on multi-objective nonlinear-programming problem with rough parameters,;
Reply: Thank you for your comment. The literature review is updated.
Comment 3: Check all equations and symbols. Some are undefined.
Reply: Checked.
Comment 4: Describe the importance of rough features in the decision-making process.
Reply: Added in the paper:
Rough numbers have recently emerged as a new concept capable of dealing with uncertain, ambiguous and probabilistic data [27,28]. This ability to deal with ambiguous and imprecise data has prompted many researchers to use these numbers in many applications and problems faced by decision makers and researchers in the real world [29]. In the same context, this paper proposes a model for the extension of interval-rough numbers.
Applying for interval rough numbers in the decision-making process enables efficient processing of imprecision and uncertainty in subjective estimates. The rough boundary interval is an imprint of uncertainty defined based on internal knowledge in the home matrix data. In cases where greater uncertainties are present in the data, the footprint of uncertainty increases, while in cases where imprecision and uncertainty from the data are eliminated, the rough number interval is transformed into a crisp value. The application of interval rough numbers eliminates the need for additional subjective assessments of boundary parameters significantly objectifies the decision-making process.
Comment 5: Add advantages of SWARA and ARAS method over the existing studies.
Reply: Please read subsection c) Adequacy to supporting group decision making as part of the fifth section and conclusion section also. Also, new text has been added:
The main advantages that determine the authors for applying the SWARA method are as follows: (1) The SWARA method has a simple and easy-to-understand mathematical apparatus. It can therefore be used effectively in group decision-making in which experts can easily exchange views; (2) The complexity of the mathematical apparatus of the SWARA method does not increase with the increase in the number of criteria. Therefore, it is suitable for application in complex mathematical models that have a number of decision attributes; and (3) the SWARA method allows the processing of data defined on the basis of different assessment scales. This feature gives this method significant flexibility, allowing easy adaptation of the SWARA model to the specific situation.
The ARAS method has proven to be a stable and reliable tool for decision-making in a dynamic environment. Compared to other traditional MCDM techniques. The ARAS method has significant resistance to the rank reversal problem. If an ideal and anti-ideal value is defined outside the range of home matrix values, the ARAS methodology algorithm eliminates the rank reversal problem. Also, numerous simulations in which larger data sets are used have shown the stability of the ARAS methodology, which determines its application in more complex studies in which larger data sets are processed in a short time.
Comment 6: Discussion of the results needs to be added in details.
Reply: The results are discussed in more details.
Comment 7: Conclusion section should be rewritten by adding the major finding, advantages and future scope of the work.
Reply: All these elements are represented through the following text:
Here, we can identify four main contributions and novelties of this study: (1) Presentation of a novel IRN SWARA-ARAS model that allows for an objective evaluation of strategies of cleaner production in an uncertain environment; (2) It has been proposed an improved MCDM methodology which is a powerful management tool for decision making; (3) The presented methodology enables the evaluation of alternatives despite the uncertainties in the decision-making process and the lack of quantitative information; (4) The IRN SWARA-ARAS model enables a flexible decision-making process and serves as a useful reference for researchers in the field of cleaner production and other operational areas.
The main limitation of the IRN SWARA-ARAS model is a complex mathematical algorithm for calculating interval rough values. The increase in the number of experts in the research further complicates the application of this algorithm. This limitation can be eliminated by creating a software solution that will be user-oriented and that will allow a wider application of the presented algorithm to solve real world problems. During the work on this study, the authors developed a software solution based on the application of MATLAB and Microsoft Excel software packages. The software solution was used to validate the IRN SWARA-ARAS model solution. Other limitations of our study include a relatively small number of criteria used to evaluate alternatives and the possible impact of survey formatting on study results.
Further research is recommended on a larger sample of respondents and considering the possibility of increasing the number of evaluation criteria through the introduction of sub-criteria within the groups of criteria. Also, further research should be directed towards the development of a universal decision-making tool based on the IRN SWARA-ARAS methodology that allows the application of a different number of criteria/sub-criteria for decision making. In addition, this study may open new directions for future research in terms of methodological [37-39] and practical application. It is recommended to focus future research on extending the proposed methodology by using hybrid fuzzy-rough numbers. This extension will allow the exploitation of the advantages of fuzzy theory and rough theory at the same time and the formation of limit values of interval fuzzy-rough numbers based on uncertainties that exist in the preferences of experts. This would eliminate the introduction of assumptions when defining the limit values of the interval of classical fuzzy numbers.
Comment 8: Correct the reference style as per the journal format. It should be uniform throughout the study.
Reply: Done. If the paper will be accepted we will additionaly improve reference style.
Comment 9: Future research work direction should be added with articles: On the Application of a Lexicographic Method to Fuzzy Linear Programming Problems; Similarity-Distance Decision-Making Technique and its Applications via Intuitionistic Fuzzy Pairs.
Reply: Thank you for your suggestion. Added.
Reviewer 2 Report
The paper is well written and structured. I make no judgement about the English. My only doubt, which the authors also introduced in the conclusions, is that the mathematical process is too complex to be used daily. Perhaps it can be employed when specific software is created. To date, the process has little applicability.
Author Response
Reviewer 2:
The paper is well written and structured. I make no judgement about the English. My only doubt, which the authors also introduced in the conclusions, is that the mathematical process is too complex to be used daily. Perhaps it can be employed when specific software is created. To date, the process has little applicability.
Thank you for your positive review. As you have mentioned we noted it in conclusion section.
The main limitation of the IRN SWARA-ARAS model is a complex mathematical algorithm for calculating interval rough values. The increase in the number of experts in the research further complicates the application of this algorithm. This limitation can be eliminated by creating a software solution that will be user-oriented and that will allow a wider application of the presented algorithm to solve real world problems. During the work on this study, the authors developed a software solution based on the application of MATLAB and Microsoft Excel software packages. The software solution was used to validate the IRN SWARA-ARAS model solution.
Reviewer 3 Report
This paper provides a a multi-criteria model of evaluation of five strategies for cleaner production based on eight sustainable criteria in Libyan industry. In order to determine the significance of the criteria, it is developed a novel interval rough stepwise weight assessment ratio analysis method, which takes into account the preferences of decision-makers by applying interval rough numbers. A novel interval rough additive ratio assessment method has been developed for the evaluation and selection of the most favourable strategy for cleaner production. The integration of the developed methods has yielded results showing that the first strategy, Launching awareness-raising campaigns to publicize these policies, represents the most realistic and best current solution to achieve cleaner production in Libyan industry. A comparative analysis with some existing interval rough methodologies have been presented to verify the superiority of the proposed model. In addition, in a sensitivity analysis, the weight of the most significant criterion has been changed.
The methods used are adapted properly and the presentation is clear. Also, the results are supported with comparative analysis.
Author Response
Reviewer 3:
This paper provides a a multi-criteria model of evaluation of five strategies for cleaner production based on eight sustainable criteria in Libyan industry. In order to determine the significance of the criteria, it is developed a novel interval rough stepwise weight assessment ratio analysis method, which takes into account the preferences of decision-makers by applying interval rough numbers. A novel interval rough additive ratio assessment method has been developed for the evaluation and selection of the most favourable strategy for cleaner production. The integration of the developed methods has yielded results showing that the first strategy, Launching awareness-raising campaigns to publicize these policies, represents the most realistic and best current solution to achieve cleaner production in Libyan industry. A comparative analysis with some existing interval rough methodologies have been presented to verify the superiority of the proposed model. In addition, in a sensitivity analysis, the weight of the most significant criterion has been changed.
The methods used are adapted properly and the presentation is clear. Also, the results are supported with comparative analysis.
__________________________________________________________________________________
Thank you for your positive review.
Round 2
Reviewer 1 Report
It is accepted now.